# Revisiting Residual Connections for Neural Structure Learning

## Abstract

Recent studies reveal that deep representation learning models without proper regularization can suffer from the dimensional collapse issue, i.e., representation vectors span over a lower dimensional space. In the domain of graph deep representation learning, the phenomenon that the node representations are indistinguishable and even shrink to a constant vector is called oversmoothing. Based on the analysis of the rank of node representations, we find that representation oversmoothing and dimensional collapse are highly related to each other for deep graph neural networks (GNNs), and the oversmoothing problem can be interpreted by the dimensional collapse of the representation matrix. Then, to address the dimensional collapse and the triggered oversmoothing in deep graph neural networks, we first find vanilla residual connections and contrastive learning producing sub-optimal outcomes by ignoring the structural information of graph data. Motivated by this, we propose a novel graph neural network named GearGNN to alleviate the oversmoothing issue from the perspective of addressing dimensional collapse in two folds. Specifically, in GearGNN, we design a topology-preserving residual connection for graph neural networks to force the low-rank of hidden representations close to the full-rank input features. Also, we propose the structure-guided contrastive loss to ensure only close neighbors who share similar local connections can have similar representations. Empirical experiments on multiple real-world datasets demonstrate that GearGNN outperforms state-of-the-art deep graph representation baseline algorithms.

## 1 Introduction

Representation learning models have achieved outstanding performance for various application domains by outputting informative hidden representations, such as computer vision and natural language processing. Recent studies (Hua et al., 2021; Jing et al., 2022; Guo et al., 2023) show that the deep representation learning models without proper regularization tend to produce representations that collapse along certain directions, known as the *dimensional collapse*, which can be further interpreted by the visualization of the singularity ranking of the matrices of representations (Hua et al., 2021). With the advent of big data, graph structures recently received increasing research attention for their ability to encode complex interactions. Similarly, the deep representation learning models on graphs are also found affected by representation issues, i.e., the node representation vectors outputted by deeper graph neural networks are not discriminative from each other and directly hurt the performance of node classification and link prediction tasks and their corresponding applications. This phenomenon is called *oversmoothing* in the graph representation learning domain (Li et al., 2018; Oono & Suzuki, 2020; Rusch et al., 2023).

In this paper, we first find that the oversmoothing in graph deep learning can be interpreted by dimensional collapse from the low-rank of representation matrix, a detailed theoretical derivation can be found in Appendix A. To empirically demonstrate the existence of dimensional collapse in the graph representation learning domain, we conduct a toy experiment on the Cora (Lu & Getoor, 2003) benchmark dataset by exploring the rank of the covariance matrix of the node representations. The analysis is visualized in Figure 1, where the $x$-axis is the index of the sorted singular values of the covariance matrix of the representation matrix, and the $y$-axis is the logarithm of the singular value. In Figure 1, we can see that the number of non-zero singular values is much smaller than the number of dimensions for a GCN graph neural network (Kipf

& Welling, 2017). This suggests that the representation matrix is *low-rank,* and the discrimination of node representation vectors only relies on a few dimensions, which naturally increases the difficulty of effectively discriminating node presentations and makes tasks like node classification and link prediction groundless.

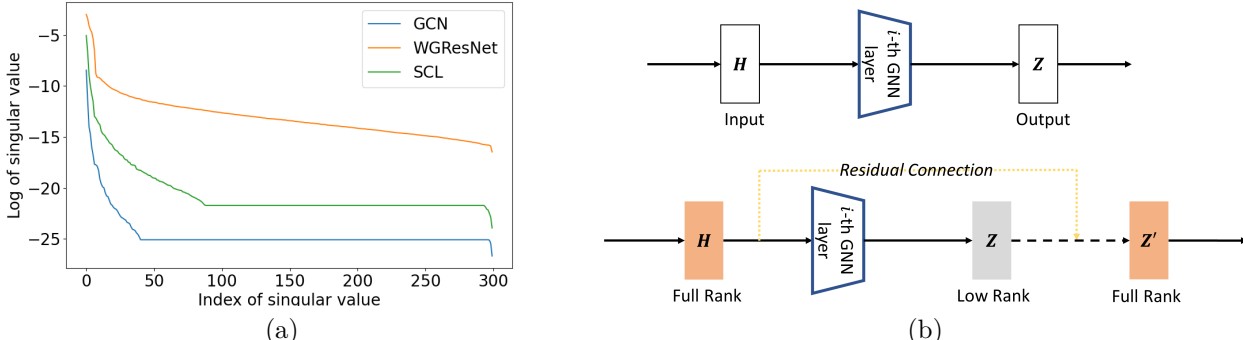

Figure 1: (a). A toy example on the Cora dataset to demonstrate the rank deficiency, where GCN is chosen as the backbone, the number of layers is set to 64 and the dimension of representation is 100. (b). The visualization about how vanilla residual connections of neural layers turn the low-rank representation $Z$ into a full-rank representation $Z'$.

To address the dimensional collapse problem in deep neural networks, residual connections (He et al., 2016) among neural layers can be an effective manner, i.e., it has been discovered that residual connections across neural network layers force the low-rank of hidden representations close to the full-rank input features (Jing et al., 2022), as shown in Figure 1 (b). The residual connections pave the way for eliminating the dimensional collapse for deep neural networks and indicate the de-oversmoothing probability for deep graph neural networks, but for non-IID graph data, we find the vanilla residual connections can produce sub-optimal results for possible two reasons. First, the vanilla residual connections ignore the topological assumption of graph data that *closer neighbors are more important during the embedding process*, simply adding residual connection can induce "*shading neighbors*" effects, i.e., even residually connected, close neighbors becomes less important during the neural representation process, as we discussed in Section 2.3. Second, targeting this specific oversmoothing phenomenon in the graph representation learning domain, the direct observation is that individual representations are indistinguishable. Hence, contrastive learning serves as a viable solution, but the existing work (Guo et al., 2023) simply introduces vanilla contrastive loss as a regularization while failing to consider the topological relationship of positive and negative pairs.

Facing the latent dimensional collapse problem (i.e., by computing the singular value of covariance matrix of representations) and observable oversmoothing problem (i.e., by discriminating node embedding vectors) in deep graph neural networks, we propose two effective directions, i.e., Weight-Decaying Graph Residual Connection (WG-ResNet) and Structure-Guided Contrastive Loss (SCL). In brief, WG-ResNet adapts weighted residual connections to preserve the input graph topology, and SCL weighs different positive and negative pairs based on their topological relations. The effectiveness of SCL and WG-ResNet in addressing dimensional collapse is also shown in Figure 1 (a). Moreover, it can be observed that SCL itself can alleviate the dimensional collapse to some extent, i.e., alleviating oversmoothing by contrastive learning addresses the dimensional collapse, which again proves the unity of dimensional collapse and oversmoothing as we discussed above.

In the end, we propose an end-to-end graph neural network model GearGNN, which encloses SCL and WG-ResNet in a GNN-agnostic manner to help arbitrary graph neural networks go deeper effectively compared to state-of-the-art baselines, supported by theoretical and empirical analysis. Furthermore, we designed extensive ablation studies to show that SCL and WG-ResNet both contribute to alleviating the dimensional collapse of the deep graph neural networks for the de-oversmoothing, and their combination can reach the optimal results.

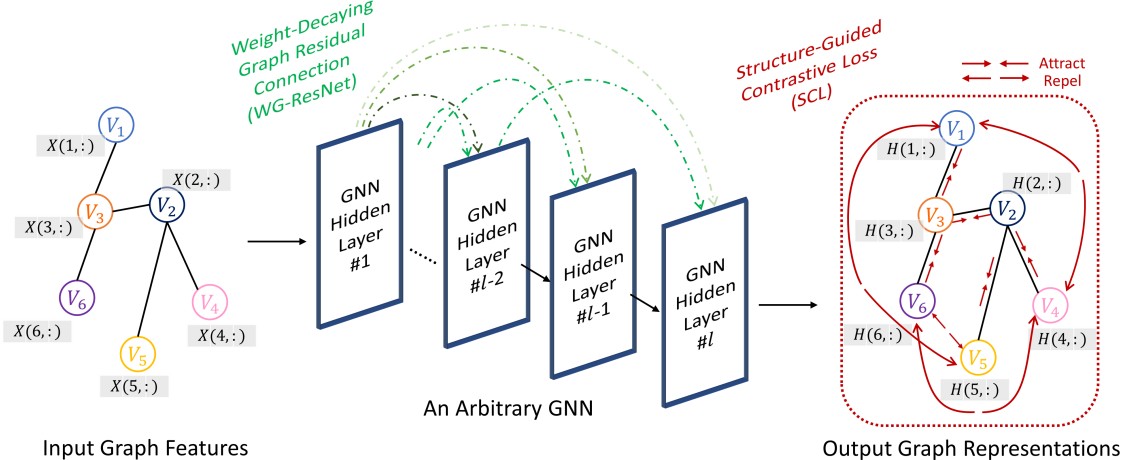

Figure 2: An arbitrary GNN with the proposed GearGNN.

## 2 Proposed Method

In this section, we begin with the overview of GearGNN and then provide the details of the Weight-decaying Graph Residual Connection (WG-ResNet) and Structure-guided Contrastive Loss (SCL). We formalize the problem of graph embedding within the context of an undirected graph $\mathcal{G} = (V, E, \boldsymbol{X})$, where $V$ consists of $n$ nodes, $E$ consists of $m$ edges, $\boldsymbol{X} \in \mathbb{R}^{n \times d}$ denotes the feature matrix and $d$ is the feature dimension. We let $\boldsymbol{A} \in \mathbb{R}^{n \times n}$ denote the adjacency matrix and denote $\boldsymbol{A}_i \in \mathbb{R}^n$ as the adjacency vector for node $v_i$. $\boldsymbol{H}_i \in \mathbb{R}^h$ is the hidden representation vector of $v_i$.

### 2.1 Overview of GearGNN

The overview of our proposed GearGNN is shown in Figure 2 and GearGNN consists of two parts, including the graph architecture WG-ResNet and contrastive loss SCL. Specifically, the green dash line stands for WG-ResNet, where $\boldsymbol{H}^{(l)}$ at the $l$-th layer will be adjusted by its second last layer $\boldsymbol{H}^{(l-2)}$ and the first layer $\boldsymbol{H}^{(1)}$ with proper weights. The red dash line in Figure 2 stands for SCL, where we first sample positive node pairs and negative node pairs based on the input graph topology such that the hidden representations of positive node pairs get closer and negative ones are pushed farther apart. The overall of GearGNN in terms of loss functions and architectures is expressed as follows.

$$\mathcal{L}_{\text{GearGNN}} = \mathcal{L}_{GNN} + \alpha \mathcal{L}_{\text{SCL}} \tag{1}$$

where $\mathcal{L}_{GNN}$ denotes the loss of the downstream task (e.g., node classification) using an arbitrary GNN model (e.g., GCN (Kipf & Welling, 2017)) equipped with WD-ResNet, $\mathcal{L}_{\text{SCL}}$ is the structure-guided contrastive loss, and $\alpha$ is a constant hyperparameter. The details of WG-ResNet and SCL are introduced below.

### 2.2 Weight-Decaying Graph Residual Connection (WG-ResNet)

As shown in Figure 1 (b), the vanilla residual connections (e.g., ResNet (He et al., 2016)) have the potential to alleviate the dimensional collapse of deep neural networks. But for deep graph neural networks, we discover that simply adding residual connections leads to the sub-optimal solution. As ResNet stacks layers, the importance of close neighbors' features gradually decreases during the GNN information aggregation process, and the faraway neighbor information becomes dominant. More concretely speaking, taking graph convolutional neural network (Kipf & Welling, 2017) as an example, the graph residual connection is expressed as follows.

$$\boldsymbol{H}^{(l)} = \sigma(\hat{\boldsymbol{A}} \boldsymbol{H}^{(l-1)} \boldsymbol{W}^{(l-1)}) + \boldsymbol{H}^{(l-2)} \tag{2}$$

where $l(\geq 2)$ denotes the index of layers, $\boldsymbol{H}^{(l-1)}$ and $\boldsymbol{H}^{(l-2)}$ are the hidden representations at corresponding layers, $\sigma(\cdot)$ is the activation function, $\boldsymbol{W}^{(l-1)}$ is the learnable weight matrix, and $\hat{\boldsymbol{A}}$ is the re-normalized self-looped adjacency matrix with $\hat{\boldsymbol{A}} = \tilde{\boldsymbol{D}}^{-\frac{1}{2}}\tilde{\boldsymbol{A}}\tilde{\boldsymbol{D}}^{-\frac{1}{2}}$ and $\tilde{\boldsymbol{A}} = \boldsymbol{A} + \boldsymbol{I}$, where $\tilde{\boldsymbol{D}}$ is the degree matrix. In ResNet, the residual connection connects the current layer and its second last layer. Without loss of generality, we assume the last layer of GNNs is stacked by ResNet, i.e., $l$ is divisible by 2. Then, by extending $\boldsymbol{H}^{(l-2)}$ iteratively (i.e., substituting it with its previous residual blocks), the above Eq. 2 could be rewritten as follows.

$$\boldsymbol{H}^{(l)} = \sigma(\hat{\boldsymbol{A}}\boldsymbol{H}^{(l-1)}\boldsymbol{W}^{(l-1)}) + \sigma(\hat{\boldsymbol{A}}\boldsymbol{H}^{(l-3)}\boldsymbol{W}^{(l-3)}) + \boldsymbol{H}^{(l-4)}$$
$$= \underbrace{\sigma(\hat{\boldsymbol{A}}\boldsymbol{H}^{(l-1)}\boldsymbol{W}^{(l-1)}) + \sigma(\hat{\boldsymbol{A}}\boldsymbol{H}^{(l-3)}\boldsymbol{W}^{(l-3)}) + \cdots}_{\text{Information aggregated from the faraway neighbors}} + \underbrace{\sigma(\hat{\boldsymbol{A}}\boldsymbol{H}^{(i)}\boldsymbol{W}^{(i)}) + \cdots + \sigma(\hat{\boldsymbol{A}}\boldsymbol{H}^{(1)}\boldsymbol{W}^{(1)})}_{\text{Information aggregated from the nearest neighbors}} \quad (3)$$

According to (Xu et al., 2019), stacking $l$ layers in GNNs and obtaining $\boldsymbol{H}^{(l)}$ can be interpreted as aggregating information from $l$-hop neighbors for node hidden representations. As shown in Eq. 3, when we stack more layers in GNNs, the information collected from faraway neighbors becomes dominant (as there are more terms regarding the information from faraway neighbors), and dilutes the information collected from the nearest neighbors (e.g., 1-hop or 2-hop neighbors). This phenomenon contradicts the general intuition that the close neighbors of a node carry the most important information, and the importance degrades with faraway neighbors. Formally, we describe this phenomenon as *shading neighbors* effect when stacking graph neural layers, as the importance of the nearest neighbors is diminishing. We empirically demonstrate that the *shading neighbors* effect degrades GNN performance in downstream tasks in Section 3.4. Specifically, we show that (1) vanilla ResNet exhibits the *shading neighbors* effect in graph representation learning; (2) jumping knowledge (Xu et al., 2018) can be a viable solution to mitigate the *shading neighbors* effect; (3) our proposed WG-ResNet achieves the best effectiveness in addressing the *shading neighbors* effect.

To formally introduce our proposed generic graph architecture, i.e., Weight-Decaying Graph Residual Connection (WG-ResNet). Here, we first introduce the formulation and then provide insights regarding why it can address the problem. Specifically, WG-ResNet introduces the layer similarity and weight decaying factor as follows.

$$\tilde{\boldsymbol{H}}^{(l)} = \sigma(\hat{\boldsymbol{A}}\boldsymbol{H}^{(l-1)}\boldsymbol{W}^{(l-1)})$$
$$\boldsymbol{H}^{(l)} = sim(\boldsymbol{H}^{(1)}, \tilde{\boldsymbol{H}}^{(l)}) \cdot e^{-l/\lambda} \cdot \tilde{\boldsymbol{H}}^{(l)} + \boldsymbol{H}^{(l-2)} \quad (4)$$
$$= e^{cos(\boldsymbol{H}^{(1)}, \tilde{\boldsymbol{H}}^{(l)}) - l/\lambda} \cdot \tilde{\boldsymbol{H}}^{(l)} + \boldsymbol{H}^{(l-2)}$$

where $cos(\boldsymbol{H}^{(1)}, \tilde{\boldsymbol{H}}^{(l)}) = \frac{1}{n}\sum_i \frac{\boldsymbol{H}_i^{(1)}(\tilde{\boldsymbol{H}}_i^{(l)})^\top}{\|\boldsymbol{H}_i^{(1)}\|\|\tilde{\boldsymbol{H}}_i^{(l)}\|}$ measures the similarity between the $l$-th layer and the 1-st layer, and we use the exponential function to map the cosine similarity ranging from $[-1, 1]$ to $[e^{-1}, e^1]$, to avoid the negative similarity weights. The term $e^{-l/\lambda}$ is the decaying factor to further adjust the similarity weight of $\tilde{\boldsymbol{H}}^{(l)}$, where $\lambda$ is a constant hyperparameter.

Different from the vanilla ResNet (He et al., 2016), we introduce a learnable similarity term $sim(\boldsymbol{H}^{(1)}, \tilde{\boldsymbol{H}}^{(l)})$ to expand the hypothesis space of deeper GNNs. As we mentioned earlier, simply adding vanilla ResNet on GNNs will cause the shading neighbors effect. The introduced decaying factor $e^{-l/\lambda}$ mitigates this effect by introducing layer-wise dependency to stacking operations and preserving hierarchical information in the graph as GNNs deepen. As $\lambda$ remains constant, the value of $e^{-l/\lambda}$ is decreasing as $l$ increases. Consequently, the later stacked layers become less influential than the previously stacked ones due to the decaying weight, effectively addressing the shading neighbors effect. In contrast, without the decaying factor, the layer-wise weights remain independent, and the shading neighbors effect persists. Moreover, we visualize the layer-wise weight distribution of different residual connection methods (including our WG-ResNet) and their effectiveness in addressing the shading neighbors effect in Appendix B. From another perspective, the hyperparameter $\lambda$ of the decaying factor actually controls the number of effective neural layers in deeper GNNs. We find its optimal value directly related to the diameter of the input graph, the detailed discussion can be found in Section 3.6.

## 2.3 Structure-Guided Contrastive Loss (SCL)

According to (Hua et al., 2021; Jing et al., 2022), contrastive representation learning methods show success in preventing dimensional collapse for image recognition whereby representation vectors shrink along certain directions. For graph structure representation learning, the contrastive methods are able to construct the positive and negative sets, and minimizing the similarity of the negative pairs allows the node representations to be uniformly distributed in the embedding space (Wang & Isola, 2020), and some nascent of contrastive learning on graphs have obtained promising alleviation in alleviating the oversmoothing issue and the corresponding oversmoothing (Zhao & Akoglu, 2020; Guo et al., 2023). However, simply adopting the idea of contrastive regularization in deep graph neural networks could not fully alleviate the oversmoothing issue due to ignoring the topological relation of non-IID graph data. Hence, mitigating oversmoothing issue in deep graph neural networks remains a significant challenge. To address this issue with the geometry consideration, we propose the Structure-guided Contrastive Loss (SCL) as follows. Intuitively, it says from an anchor node say $v_i$, its positive sample (i.e., connected node) should have close node representations, and its negative sample (i.e., disconnected node) should have discriminative node representations. Also, $sigma$ and $\gamma$ measures the importance of positive and negative samples based on the input topology.

$$\mathcal{L}_{\text{SCL}} = -\mathbb{E}_{v_i \in V}[\mathbb{E}_{v_j \in \mathcal{N}_i}(\sigma_{ij} \log(f(\boldsymbol{z}_i, \boldsymbol{z}_j))) + \mathbb{E}_{v_k \in \bar{\mathcal{N}}_i}(\gamma_{ik} \log(1 - f(\boldsymbol{z}_i, \boldsymbol{z}_k)))]$$

$$\sigma_{ij} = \frac{n^2}{m} \cdot \frac{1 - dist(\boldsymbol{A}_i, \boldsymbol{A}_j)/n}{\sum_{v_{i'} \in V, v_{j'} \in \mathcal{N}_{i'}}(1 - dist(\boldsymbol{A}_{i'}, \boldsymbol{A}_{j'})/n)}, \gamma_{ik} = \frac{n^2}{n^2 - m} \cdot \frac{1 + dist(\boldsymbol{A}_i, \boldsymbol{A}_k)/n}{\sum_{v_{i'} \in V, v_{k'} \in \bar{\mathcal{N}}_{i'}}(1 + dist(\boldsymbol{A}_{i'}, \boldsymbol{A}_{k'})/n)}$$

(5)

where $\boldsymbol{z}_i = g(\boldsymbol{H}_i^{(l)})$, $g(\cdot)$ is an encoder mapping $\boldsymbol{H}_i^{(l)}$ to another latent space, $f(\cdot)$ is a similarity function (e.g., $f(a, b) = \exp(\frac{ab^\top}{||a||||b||})$), $dist(\cdot)$ is a distance measurement function (e.g., hamming distance (Norouzi et al., 2012)), $\mathcal{N}_i$ is the set containing one-hop neighbors of node $v_i$, $\bar{\mathcal{N}}_i$ is the complement of the set $\mathcal{N}_i$, $m$ is the number of edges and $n$ is the number of vertices. In Eq. 5, the connected edge $(v_i, v_j)$ forms the positive pair, while disconnected edge $(v_i, v_k)$ forms the negative pair. Moreover, $(v_{i'}, v_{j'})$ iterates over all connected edges in the input graph, and $(v_{i'}, v_{k'})$ iterates over all disconnected edges in the input graph.

The intuition of Eq. 5 is to maximize the similarity of the representations of the positive pairs and to minimize the similarity of the representations of the negative pairs, such that the node representations become discriminative. In which process, some research works (Perozzi et al., 2014; Grover & Leskovec, 2016; Le, 2021) would first assume that the importance of each edge is identical. However, such an assumption does not always get satisfied in many applications (Velickovic et al., 2017; Faisal et al., 2015). To address this issue, we reweight the importance of edges by considering the graph topological structure via $\sigma$ and $\gamma$. Therefore, for a positive pair, if two nodes have similar topological structures, the weight (i.e., $\sigma$) of this node pair should be large; for a negative pair, if two nodes have similar topological structures, the weight (i.e., $\gamma$) of this node pair should be small.

Next, we show the importance of these topology-aware weights (i.e., $\sigma$ and $\gamma$) in mitigating the oversmoothing issue with the theoretical analysis. For delivering the following analysis clearly, we adopt node $v_i$ as the anchor node for illustration, without loss of generality (Zhu et al., 2021) Mathematically, some works (Perozzi et al., 2014; Grover & Leskovec, 2016; Le, 2021) assume the importance of each edge is identical, i.e., the edge distribution $P(e)$ is uniform. Consequently, our objective is to initially recalibrate the importance of edges by incorporating the topological information of the graph. Subsequently, we aim to present the ensuing topology-aware distribution. Note that, in the following derivation, the edge $e$ we referred to consists of existing edges and non-existing edges. To be specific, if node $i$ and node $k$ do not connect in the input graph, then $e_{ik}$ represents the non-existing edge between node $i$ and node $k$, denoted as a negative edge.

To begin with, we denote the probability of sampling a connection $e_{ij}$ and it is a positive connection as $\tilde{P}_{pos}(e_{ij}) \propto P(e_{ij}, y = 1)$, i.e., the sampled pair of two nodes $v_i$ and $v_j$ connect in the input graph. Then, $\tilde{P}_{pos}(e_{ij})$ can be further expressed as $\tilde{P}_{pos}(e_{ij}) = \sigma_{ij}P_{pos}(e_{ij})$, where $P_{pos}(e_{ij}) = \frac{m}{n^2}$ is the prior probability that the sampling is positive, and $\sigma_{ij}$ is the conditional probability for the joint probability $\tilde{P}_{pos}(e_{ij})$. Note that a positive connection (e.g., $e_{ij}$) stands for two connected nodes $v_i$ and $v_j$ forming a positive pair. Similarly, for disconnected two nodes $v_i$ and $v_k$ (i.e., the negative pair or negative connection $e_{ik}$), we denote

$\tilde{P}_{neg}(e_{ik}) = \gamma_{ik}P_{neg}(e_{ik})$, where $P_{neg}(e_{ik}) = 1 - \frac{m}{n^2}$ is the prior probability of sampling a negative connection, and we interpret $\gamma_{ik}$ as the conditional probability for the joint probability $\tilde{P}_{neg}(e_{ik})$.

Finally, we denote $\theta$ to be the parameters of the multi-layer GNN model $\mathcal{G}_\theta(\cdot)$, i.e., $Z = \mathcal{G}_\theta(A, X)$, such that we can prove that SCL could alleviate the oversmoothing issue from the perspective of generative adversarial network (GAN) (Goodfellow et al., 2014) as follows.

**Proposition 2.1.** $\mathcal{L}_{SCL}$, *based on contrastive learning, can be interpreted as the objective function of a generative adversarial network (GAN) (Goodfellow et al., 2014), which could be written as follows.*

$$\min_\theta \mathcal{L}_{SCL} = \max_\theta \int_e (\tilde{P}_{pos}(e)\log(D(e)) + \tilde{P}_{neg}(e)\log(1 - D(e)))de$$

*where $D(e) = f(\boldsymbol{z}_i, \boldsymbol{z}_j)$ is the discriminator of GAN with edge $e = (v_i, v_j)$, node representations $\boldsymbol{z}_i$ and $\boldsymbol{z}_j$. Probabilities $\tilde{P}_{pos}(e) = \sigma P_{pos}(e)$ and $\tilde{P}_{neg}(e) = \gamma P_{neg}(e)$ are defined above. (Proof in Appendix C)*

According to Proposition 2.1, the proposed $\mathcal{L}_{SCL}$ can be interpreted as distinguishing the existence of a certain edge based on the representation vectors of two nodes. Next, we then derive how this design helps, as a regularizer, to alleviate the oversmoothing problem based on positive and negative samples.

**Proposition 2.2.** *The regularization term $\mathcal{L}_{SCL}$ can alleviate the oversmoothing problem (i.e., the node hidden representation vectors are distinguishable), given the optimal discriminator in Proposition 2.1.*

*Proof.* Based on Proposition 2.1 and following Theorem 1 in GAN (Goodfellow et al., 2014), the optimal $D^*(e)$ can be derived and denoted as $D^*(e) = \frac{\tilde{P}_{pos}(e)}{\tilde{P}_{pos}(e)+\tilde{P}_{neg}(e)}$. Since $\tilde{P}_{pos}(e_{ij}) = \sigma_{ij}P_{pos}(e_{ij})$ and $\tilde{P}_{neg}(e_{ik}) = \gamma_{ik}P_{neg}(e_{ik})$, we have

$$\begin{aligned}
D^*(e) &= \frac{\sigma * P_{pos}(e)}{\sigma * P_{pos}(e) + \gamma * P_{neg}(e)} \\
&= \frac{P(e|y=1)P(y=1)}{P(e|y=1)P(y=1) + P(e|y=0)P(y=0)} \\
&= P(y=1|e)
\end{aligned}$$

Therefore, $D(e)$ can be interpreted as maximizing the conditional log-likelihood $P(y = 1|e)$, where $y = 1$ indicates edge $e$ is a positive (i.e., exiting) edge. Notice that the discriminator $D(e)$ is defined as the similarity measurement of a node pair in terms of their representation vectors, as stated in Proposition 2.1. In other words, if $D(e)$ is able to distinguish whether a node pair is a negative pair or not, the hidden representations of these two nodes (negative pair) are then distinguishable. Thus, *we can conclude that when $\mathcal{L}_{SCL}$ achieves the optimal solution, the model could successfully discriminate the difference of the hidden representation vectors for the negative pairs,* thus alleviating the oversmoothing issue. In practice, the optimum is usually approximated by the model convergence. □

## 3 Experiments

In this section, we comprehensively demonstrate the performance of our proposed GearGNN compared to state-of-the-art deeper graph neural networks and self-ablations, trying to answer the following research questions.

- RQ1: When do we need more layers of graph neural networks? *(Answered in Section 3.2)*

- RQ2: When it is necessary to be deep, can the proposed GearGNN alleviate dimensional collapse and oversmoothing to outperform? *(Answered in Section 3.3)*

- RQ3: Is every component of GearGNN helpful and irreplaceable? *(Answered in Section 3.4)*

- RQ4: In practice, can GearGNN be agnostic to help various off-the-shelf graph neural network architectures? *(Answered in Section 3.5)*

### 3.1 Experiment Setup

**Datasets**. *Cora* (Lu & Getoor, 2003) dataset is a citation network consisting of 5,429 edges and 2,708 scientific publications from 7 classes. The edge in the graph represents the citation of one paper by another. *CiteSeer* (Lu & Getoor, 2003) dataset consists of 3,327 scientific publications which could be categorized into 6 classes, and this citation network has 9,228 edges. *PubMed* (Namata et al., 2012) is a citation network consisting of 88,651 edges and 19,717 scientific publications from 3 classes. *Reddit* (Hamilton et al., 2017b) dataset is extracted from Reddit posts, which consists of 4,584 nodes and 19,460 edges. Notice that we follow the splitting strategy used in (Zhao & Akoglu, 2020) by randomly sampling 3% of the nodes as the training samples, 10% of the nodes as the validation samples, and the remaining 87% as the test samples. *OGB-arXiv* (Wang et al., 2020) is a citation network, which consists of 1,166,243 edges and 169,343 nodes from 40 classes.

**Baselines**. We compare the performance of our method with the following baselines including one vanilla GNN model and four state-of-the-art deeper GNN models: (1) GCN (Kipf & Welling, 2017): the vanilla graph convolutional network; (2) GCNII (Chen et al., 2020): an extension of GCN with skip connections and additional identity matrices; (3) DGN (Zhou et al., 2020): the differentiable group normalization for GNNs to normalize nodes within the same group and separate nodes among different groups; (4) PairNorm (Zhao & Akoglu, 2020): a GNN normalization layer designed to prevent node representations from becoming too similar; (5) DropEdge (Rong et al., 2020): a GNN-agnostic framework that randomly removes a certain number of edges from the input graph; (6) RevGCN-Deep (Li et al., 2021): equilibrium model based deep graph neural networks; (7) EGNN (Zhou et al., 2021): dirichlet energy constrained deep graph neural networks; (8) ContraNrom (Guo et al., 2023): a contrastive learning-based layer normalization method.

**Configurations**. For a fair comparison, we set the dropout rate to 0.5, the weight decay rate to 0.0005, and the total number of iterations to 1500 for all methods; if not specialized, GCN is chosen as the backbone, and the dimension of each layer is set to 50 for all the graph neural network baseline methods. In Section 3.4, for GearGNN and GearGNN-S, we sample 10 instances and 5 neighbors for each class from the training set, $dist(\cdot)$ is the hamming distance, and $f(\cdot)$ is the cosine similarity measurement. The experiments are repeated 10 times if not otherwise specified. All of the real-world datasets are publicly available. The experiments are performed on a Windows machine with a 16GB RTX 5000 GPU. Detailed reproducibility with released code can be found in Appendix D.

### 3.2 When do we need more layers of graph neural networks?

**Case 1: Missing Features**. We first consider a scenario where some attribute values are missing in the input graph. In such cases, shallow GNNs may not perform well because they cannot gather useful information from neighbors due to the presence of numerous missing values. However, by increasing the number of layers, GNNs can gather more information from $k$-hop neighbors and capture latent knowledge to compensate for missing features. To verify this, we conducted the following experiment: we randomly masked $p\%$ of attributes on the Cora and CiteSeer datasets (i.e., setting the masked attributes to 0), gradually increased the number of layers, and recorded the accuracy for each setting following the approach in (Zhao & Akoglu, 2020). In this case study, we selected the number of layers from the set $\{2, 3, 4, 5, 6, 7, 8, 9, 10, 15, 20, 25, 30, 40, 50, 60\}$, and the backbone model used was GCN. For a fair comparison, we added ResNet (He et al., 2016) if it could enhance the baseline model's performance. We repeated the experiments five times and recorded the mean accuracy and standard deviation.

Table 1 shows the performance of GearGNN and various baselines with the optimal number of layers denoted as #L, i.e., when the model achieves the best performance. By observation, we find that when the missing rate is 25%, shallow GCN with ResNet has enough capacity to achieve the best performance on both CiteSeer and Cora datasets. Compared to GCN, our proposed method further improves the performance by more than 3.83% on the CiterSeer dataset and 4.08% on the Cora dataset by stacking more layers. However, when we increase the missing rate to 50% and 75%, we observe that most methods tend to achieve the best performance by stacking more layers. Specifically, PairNorm achieves the best performance at 10 layers when 25% features are missing, while it has the best performance at 40 layers when 75% features are missing. A similar observation could also be found with GCNII on the Cora dataset, DropEdge on the CiteSeer dataset as well as our proposed methods in both datasets. Overall, the experimental results verify that the more features a dataset are missing, the more layers GNNs need to be stacked to achieve better performance. Our

Table 1: Node Classification on Two Datasets by Masking $p\%$ of Input Node Attributes ($L$ denotes the number of layers where a model achieves the best performance).

| Node Feature Missing Rate | | $p = 25\%$ | | $p = 50\%$ | | $p = 75\%$ | |
|---|---|---|---|---|---|---|---|
| dataset | Method | Accuracy | $L$ | Accuracy | $L$ | Accuracy | $L$ |
| Cora | GCN + ResNet | $0.7503 \pm 0.0101$ | 7 | $0.7435 \pm 0.0048$ | 10 | $0.7226 \pm 0.0099$ | 10 |
| | PairNorm + ResNet | $0.7529 \pm 0.0129$ | 10 | $0.7482 \pm 0.0172$ | 20 | $0.7262 \pm 0.0178$ | 40 |
| | DropEdge + ResNet | $0.7634 \pm 0.0112$ | 15 | $0.7611 \pm 0.0102$ | 20 | $0.7297 \pm 0.0168$ | 8 |
| | GCNII + ResNet | $0.2667 \pm 0.0063$ | 25 | $0.3351 \pm 0.0066$ | 25 | $0.2914 \pm 0.0106$ | 40 |
| | DGN w/o ResNet | $0.6850 \pm 0.0184$ | 30 | $0.6846 \pm 0.0147$ | 50 | $0.6717 \pm 0.0156$ | 25 |
| | ContraNorm + ResNet | $0.7319 \pm 0.0099$ | 2 | $0.7189 \pm 0.0091$ | 3 | $0.6902 \pm 0.0107$ | 3 |
| | GearGNN | $\mathbf{0.7915 \pm 0.0060}$ | 10 | $\mathbf{0.7848 \pm 0.0043}$ | 20 | $\mathbf{0.7598 \pm 0.0081}$ | 60 |
| CiteSeer | GCN + ResNet | $0.6141 \pm 0.0080$ | 4 | $0.5811 \pm 0.0093$ | 10 | $0.5149 \pm 0.0173$ | 9 |
| | PairNorm + ResNet | $0.6184 \pm 0.0087$ | 8 | $0.5947 \pm 0.0083$ | 20 | $0.5176 \pm 0.0075$ | 10 |
| | DropEdge + ResNet | $0.6348 \pm 0.0156$ | 4 | $0.6083 \pm 0.0128$ | 6 | $0.5240 \pm 0.0128$ | 10 |
| | GCNII + ResNet | $0.2453 \pm 0.0045$ | 40 | $0.2338 \pm 0.0028$ | 20 | $0.2403 \pm 0.0046$ | 25 |
| | DGN w/o ResNet | $0.4560 \pm 0.0162$ | 20 | $0.4593 \pm 0.0117$ | 15 | $0.4498 \pm 0.0292$ | 15 |
| | ContraNorm + ResNet | $0.5893 \pm 0.0114$ | 2 | $0.5621 \pm 0.0111$ | 3 | $0.4646 \pm 0.0076$ | 4 |
| | GearGNN | $\mathbf{0.6524 \pm 0.0087}$ | 20 | $\mathbf{0.6169 \pm 0.0063}$ | 60 | $\mathbf{0.5576 \pm 0.0070}$ | 50 |

explanation for this observation is that if the number of layers increases, more information will be collected from the $k$-hop neighbors to recover the missing information of its 1-hop and 2-hop neighbors.

**Case 2: Disalignment of topological and feature distribution**. We then conducted another case study using a toy example to demonstrate that nearby neighbors may not necessarily share similar contents in terms of input features. Initially, we utilized an existing package (specifically, the draw circle function in the Scikit-learn package) to generate a synthetic dataset with 1,000 data points and a noise level set to 0.01. Subsequently, we computed the Euclidean distance between each pair of data points. If the distance between two data points was less than a predefined threshold, we connected them in a graph, resulting in the derivation of the adjacency matrix with added self-loops. Following this, we randomly sampled 1% of the data points as the training set, 9% as the validation set, and 90% as the test set. These data points were visualized in Figure 3a, and the corresponding experimental results are presented in Figure 3b. In Figure 3a, we observed that the query node (depicted as the blue diamond within the dashed circle) could not rely solely on its closest labeled neighbor (the red star within the dashed circle) to predict its label (red or blue) correctly. Only by exploring longer paths consisting of more similar neighbors were we able to accurately predict its label as blue (as indicated by the blue star within the dashed circle). Figure 3b compares the classification accuracy of shallow GNNs with that of deeper GNNs. Notably, deeper GNNs exhibited a significant performance improvement of over 11% compared to shallow ones, contributing to their capability of exploring longer paths within the graph.

### 3.3 Effectiveness Analysis of GearGNN

Here, we evaluate the effectiveness of the proposed method on benchmark datasets by comparing it with state-of-the-art methods shown in Table 2. The backbone model for all methods we used in these experiments is GCN (Kipf & Welling, 2017). For a fair comparison, we set the dimension of the hidden layer to 50 and vary the number of hidden layers from 2 to 16, 32, and 64 for all methods on the small dataset. Additionally, we examine the node classification performance of GearGNN on the large-scale dataset OGB-arXiv, as detailed in Table 3. In OGB-arXiv, we fix the feature dimension of the hidden layer as 100, set the total iteration to 3000, and choose GCN as the backbone model. Due to memory limitations, we only record the performance of all methods by setting the number of layers to 2, 10, and 20, respectively. The experiments are repeated 5 times, and we record the mean accuracy as well as the standard deviation.

Based on the observation in Table 2 and Table 3, we find that (1) many existing graph de-oversmoothing methods (e.g., PairNorm, ContraNorm) achieve the best performance with the shallow layer (i.e., $L = 2$), and their performance begin to decrease as the number of layers increases; (2) GearGNN outperforms all baseline methods when stacking the layers of GNN (i.e., $L = 16$, $L = 32$ and $L = 64$); (3) when comparing with a 64-layer GCN as the reference, DropEdge, DropEdge, and ContraNorm show worse performance than

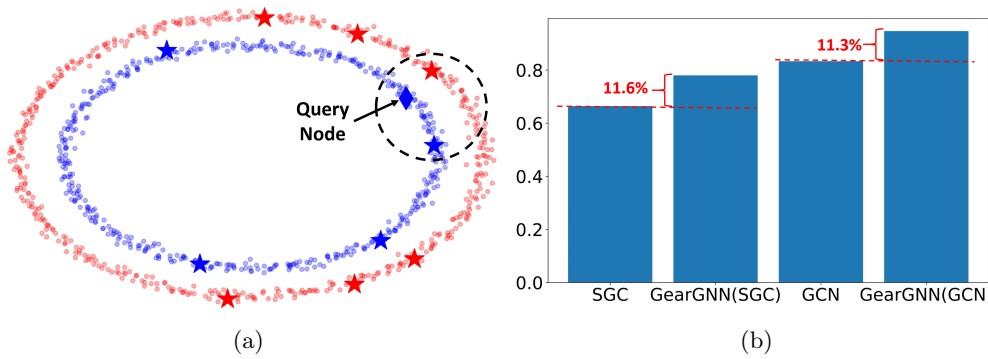

(a)                                                                (b)

Figure 3: A Toy Example to Demonstrate the Benefit of Deeper GNN Models. (a) Two groups of nodes in the semi-supervised setting. Stars are labeled, dots are unlabeled, and the diamond is the query node. Euclidean distance between two nodes indicates the edge connection. (b) Comparison of node classification accuracy between shallow and deeper GNN models using data on the left. The deeper GNNs are realized by GearGNN with corresponding backbones.

Table 2: Node Classification on Small Datasets with Varying Layers $L$ (GCN as the Backbone).

| Dataset | Method | $L = 2$ | $L = 16$ | $L = 32$ | $L = 64$ |
|---|---|---|---|---|---|
| Cora | GCN | $0.7643 \pm 0.0040$ | $0.5262 \pm 0.0732$ | $0.3284 \pm 0.0066$ | $0.3274 \pm 0.0189$ |
| | PairNorm | $0.7818 \pm 0.0027$ | $0.6080 \pm 0.0310$ | $0.5138 \pm 0.0299$ | $0.2932 \pm 0.0120$ |
| | DropEdge | $\mathbf{0.7828 \pm 0.0075}$ | $0.7557 \pm 0.0072$ | $0.7306 \pm 0.0134$ | $0.2685 \pm 0.0647$ |
| | GCNII | $0.6778 \pm 0.0065$ | $0.7237 \pm 0.0055$ | $0.7142 \pm 0.0015$ | $0.7107 \pm 0.0047$ |
| | DGN | $0.7545 \pm 0.0003$ | $0.6785 \pm 0.0169$ | $0.7067 \pm 0.0190$ | $0.7104 \pm 0.0192$ |
| | ContraNorm | $0.7682 \pm 0.0044$ | $0.6590 \pm 0.0291$ | $0.5128 \pm 0.0241$ | $0.4328 \pm 0.0320$ |
| | GearGNN | $0.7768 \pm 0.0057$ | $\mathbf{0.8002 \pm 0.0058}$ | $\mathbf{0.7961 \pm 0.0055}$ | $\mathbf{0.8022 \pm 0.0061}$ |
| CiteSeer | GCN | $0.6452 \pm 0.0072$ | $0.4514 \pm 0.0987$ | $0.2689 \pm 0.0099$ | $0.2680 \pm 0.0093$ |
| | PairNorm | $0.6030 \pm 0.0153$ | $0.2268 \pm 0.0398$ | $0.2096 \pm 0.0029$ | $0.2076 \pm 0.0033$ |
| | DropEdge | $0.6532 \pm 0.0068$ | $0.6117 \pm 0.0229$ | $0.5101 \pm 0.0430$ | $0.2138 \pm 0.0198$ |
| | GCNII | $0.5912 \pm 0.0106$ | $0.6180 \pm 0.0031$ | $0.6159 \pm 0.0019$ | $0.6101 \pm 0.0017$ |
| | DGN | $0.4872 \pm 0.0168$ | $0.4753 \pm 0.0591$ | $0.4604 \pm 0.0162$ | $0.4417 \pm 0.0219$ |
| | ContraNorm | $0.6263 \pm 0.0061$ | $0.4621 \pm 0.0237$ | $0.3965 \pm 0.0196$ | $0.2128 \pm 0.0208$ |
| | GearGNN | $\mathbf{0.6577 \pm 0.0065}$ | $\mathbf{0.6650 \pm 0.0059}$ | $\mathbf{0.6655 \pm 0.0031}$ | $\mathbf{0.6685 \pm 0.0066}$ |
| PubMed | GCN | $0.7990 \pm 0.0017$ | $0.5383 \pm 0.0200$ | $0.5463 \pm 0.0391$ | $0.5566 \pm 0.0086$ |
| | PairNorm | $0.8120 \pm 0.0076$ | $0.4408 \pm 0.0683$ | $0.3972 \pm 0.0094$ | $0.3960 \pm 0.0097$ |
| | DropEdge | $0.8035 \pm 0.0020$ | $0.7893 \pm 0.0042$ | $0.7902 \pm 0.0032$ | $0.3951 \pm 0.0108$ |
| | GCNII | $0.8070 \pm 0.0009$ | $0.8094 \pm 0.0010$ | $0.8089 \pm 0.0007$ | $0.8097 \pm 0.0009$ |
| | DGN | $0.7947 \pm 0.0358$ | $0.7553 \pm 0.0295$ | $0.7733 \pm 0.0143$ | $0.7632 \pm 0.0226$ |
| | ContraNorm | $0.8061 \pm 0.0020$ | $0.5672 \pm 0.0684$ | $0.4348 \pm 0.0379$ | $0.3971 \pm 0.0057$ |
| | GearGNN | $\mathbf{0.8175 \pm 0.0016}$ | $\mathbf{0.8097 \pm 0.0038}$ | $\mathbf{0.8098 \pm 0.0025}$ | $\mathbf{0.8109 \pm 0.0033}$ |
| Reddit | GCN | $0.8757 \pm 0.0054$ | $0.8540 \pm 0.0451$ | $0.3655 \pm 0.0251$ | $0.3410 \pm 0.0288$ |
| | PairNorm | $0.7704 \pm 0.0052$ | $0.8636 \pm 0.0448$ | $0.6468 \pm 0.0429$ | $0.1230 \pm 0.0299$ |
| | DropEdge | $0.8564 \pm 0.0059$ | $0.8526 \pm 0.0046$ | $0.5384 \pm 0.1049$ | $0.1053 \pm 0.0148$ |
| | GCNII | $0.6184 \pm 0.0108$ | $0.7157 \pm 0.0016$ | $0.6972 \pm 0.0039$ | $0.6963 \pm 0.0059$ |
| | DGN | $0.7829 \pm 0.0137$ | $0.7397 \pm 0.0371$ | $0.6806 \pm 0.0639$ | $0.5058 \pm 0.0754$ |
| | ContraNorm | $0.6576 \pm 0.0094$ | $0.2563 \pm 0.0091$ | $0.2547 \pm 0.0170$ | $0.2664 \pm 0.0140$ |
| | GearGNN | $\mathbf{0.8762 \pm 0.0060}$ | $\mathbf{0.9676 \pm 0.0033}$ | $\mathbf{0.9693 \pm 0.0023}$ | $\mathbf{0.9721 \pm 0.0011}$ |

the vanilla GCN; (4) GCNII, DGN, and GearGNN exhibit an average performance boost of more than 150% compared to GCN; (5) GCNII and GearGNN perform better with deep graph architecture (e.g., $L = 32$ or $L = 64$); (6) the performance of GearGNN on OGB-arXiv dataset increases as we stack layers, which verifies our hypothesis that increasing the number of layers indeed leads to better performance in large graphs due to more information aggregated from neighbors; (7) setting the number of layers to 10 for the OGB-arXiv dataset results in a drop in performance for most baseline methods, which deteriorates rapidly with further layer stacking. Additional comparison with RevGCN-Deep and EGNN can be found in Appendix E.

Table 3: Node Classification on Large Dataset with Varying Layers $L$ (GCN as the Backbone).

| Dataset | Method | $L = 2$ | $L = 10$ | $L = 20$ |
|---|---|---|---|---|
| OGB-arXiv | GCN | $0.7136 \pm 0.0044$ | $0.7021 \pm 0.0018$ | $0.5377 \pm 0.0756$ |
| | PairNorm | $0.7186 \pm 0.0008$ | $0.7158 \pm 0.0035$ | $0.5796 \pm 0.0090$ |
| | DropEdge | $0.7178 \pm 0.0012$ | $0.6531 \pm 0.0056$ | $0.2198 \pm 0.0097$ |
| | GCNII | $0.5966 \pm 0.0013$ | $0.6340 \pm 0.0017$ | $0.6246 \pm 0.0015$ |
| | DGN | $0.6039 \pm 0.0037$ | $0.5746 \pm 0.0033$ | $0.5027 \pm 0.0056$ |
| | ContraNorm | $0.7294 \pm 0.0025$ | $0.6941 \pm 0.0030$ | $0.5821 \pm 0.0324$ |
| | GearGNN | $\mathbf{0.7369 \pm 0.0014}$ | $\mathbf{0.7386 \pm 0.0006}$ | $\mathbf{0.7401 \pm 0.0009}$ |

In addition to Table 2 and Table 3, we visualize the corresponding number of the nonzero singular values on those datasets in Figure 4. In Figure 4, taking Cora and OGB-arXiv as examples, we observe that (1) PairNorm and ContraNorm begin to suffer from the dimensional collapse issue on both datasets when the number of layers is greater than 10; (2) Dropedge, DGN, and GCNII perform well on the small dataset but fail to preserve the full-rank representation on the large dataset; (3) node representations by GearGNN are full-rank on both datasets, indicating that GearGNN effectively alleviates the dimensional collapse.

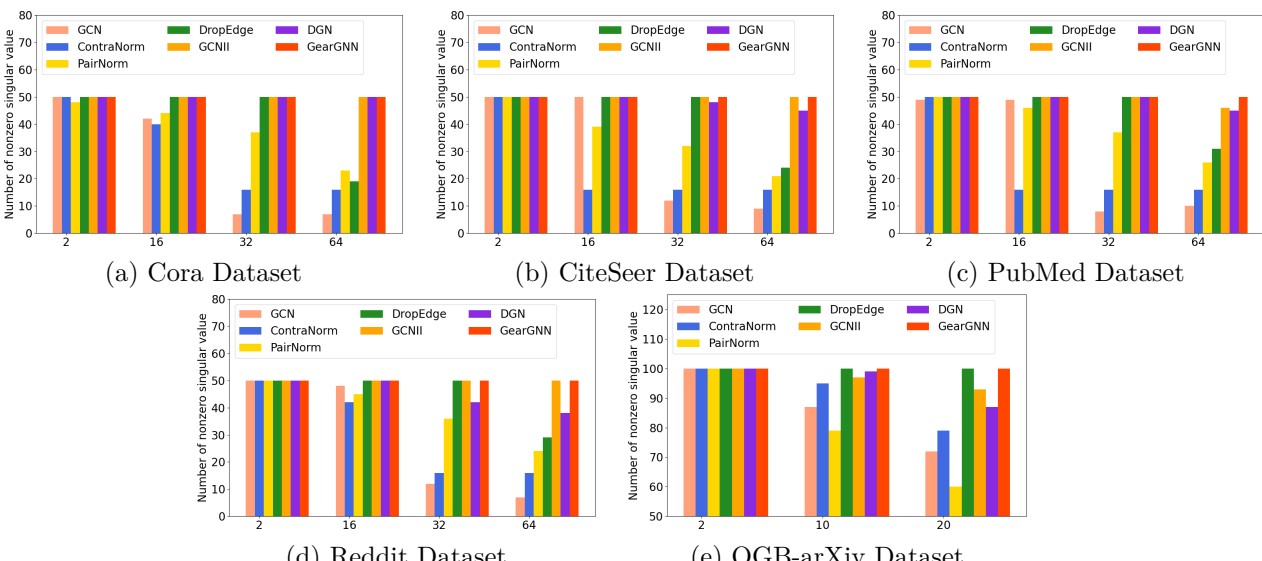

(a) Cora Dataset      (b) CiteSeer Dataset      (c) PubMed Dataset

(d) Reddit Dataset      (e) OGB-arXiv Dataset

Figure 4: The $x$-axis is the number of layers and the $y$-axis is the number of the non-zero singular values of the covariance matrix of the node representations by different methods.

Combined with the observation in Figure 4, Table 2 and Table 3, most of the baseline methods suffer from the dimensional collapse issue and not fully address the oversmoothing problem when we stack graph neural network layers, while our proposed GearGNN could largely alleviate the dimensional collapse issue in terms of both node classification performance and the singularity ranking.

## 3.4 Ablation Study of GearGNN

Here, we conduct the ablation study to show the effectiveness and irreplaceability of WG-ResNet and SCL in terms of node classification in Table 4. In this experiment, we fix the total iteration set as 3000, and GCN is chosen as the backbone model. For the Cora dataset, the feature dimension of the hidden layer is 50 and the number of layers is 64; for the OGB-arXiv dataset the feature dimension of the hidden layer is 100 and the number of layers is 20. In Table 4, GearGNN-T removes SCL, GearGNN-D removes the weight decaying factor in WG-ResNet and GearGNN-JK replaces the WG-ResNet by Jumping Knowledge (Xu et al., 2018).

Table 4: Ablation Study w.r.t. Node Classification Accuracy.

| Method | Cora ($L = 64$) | CiteSeer ($L = 64$) | PubMed ($L = 64$) | Reddit ($L = 64$) | OGB-arXiv ($L = 20$) |
|---|---|---|---|---|---|
| GCN+RseNet | $0.7252 \pm 0.0176$ | $0.6213 \pm 0.0056$ | $0.7985 \pm 0.0068$ | $0.9432 \pm 0.0037$ | $0.7144 \pm 0.0013$ |
| GearGNN-D | $0.7498 \pm 0.0139$ | $0.6567 \pm 0.0052$ | $0.8050 \pm 0.0031$ | $0.9654 \pm 0.0028$ | $0.7363 \pm 0.0011$ |
| GearGNN-T | $0.7875 \pm 0.0092$ | $0.5750 \pm 0.0244$ | $0.8078 \pm 0.0047$ | $0.9397 \pm 0.0042$ | $0.7335 \pm 0.0024$ |
| GearGNN-JK | $0.7955 \pm 0.0078$ | $0.6600 \pm 0.0085$ | $0.8061 \pm 0.0038$ | $0.9659 \pm 0.0046$ | $0.7368 \pm 0.0012$ |
| GearGNN | $\mathbf{0.8022 \pm 0.0061}$ | $\mathbf{0.6685 \pm 0.0066}$ | $\mathbf{0.8109 \pm 0.0033}$ | $\mathbf{0.9721 \pm 0.0011}$ | $\mathbf{0.7401 \pm 0.0009}$ |

In Table 4, we have the following observations (1) comparing GearGNN with GearGNN-T, we find that GearGNN boosts the performance by 1.84% on the Cora dataset after adding SCL, which demonstrates the effectiveness of SCL to alleviate the oversmoothing issue; (2) GearGNN outperforms GearGNN-D on Cora dataset by 5.61%, which shows that GearGNN could alleviate the shading neighbors effect by adding the weight decaying factor; (3) comparing GearGNN with GearGNN-JK, we verify that our proposed WG-ResNet is more effective than GearGNN-JK. Besides, one drawback of jumping knowledge is its high memory required as the number of layers increases, while our proposed WG-ResNet doesn't; (4) GearGNN outperforms GCN+ResNet by more than 7.7% on the Cora dataset and 2.6% on the OGB-arXiv dataset, which indicates that WG-ResNet could alleviate the shading neighbors effect.

### 3.5 Different Backbones of GearGNN

Here, we show the performance of our proposed GearGNN cooperating with different backbone models (e.g., GAT (Velickovic et al., 2018) and GraphSage (Hamilton et al., 2017a)). In Figure 5, we set the numbers of the hidden layers as 60 for all methods and the dimension of the hidden layer as 50. The total number of training iterations is 1500.

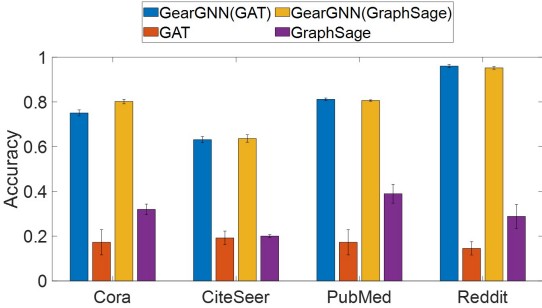

Figure 5: Accuracy of Different Backbone Models with 64 Hidden Layers on Four Datasets.

By observation, we find that both GAT and GraphSage tend to have worse performance when the architecture becomes deeper, and our proposed method GearGNN greatly boosts the performance by 40%-60% on average over four datasets. Specifically, compared with the vanilla GraphSage, our GearGNN boosts its performance by 43% on the CiteSeer dataset and more than 67% on the Reddit dataset.

### 3.6 Number of Effective Layers in Deep Graph Neural Networks

We conduct the hyperparameter analysis of GearGNN, regarding $\lambda$ in the weight decaying function of Eq. 4. For example, when $\lambda = 10$, the decaying factor for the 10-th layer is 0.3679 (i.e., $e^{-1}$); but for the 30-th layer, it is 0.0049 (i.e., $e^{-3}$). This decay limits the effective information aggregation scope of deeper GNNs because the later stacked layers will become significantly less important. Based on this controlling property of $\lambda$, a natural follow-up question is whether its value depends on the property of input graphs.

Interestingly, through our experiments, we find that *the optimal $\lambda$ is very close to the diameter of input graphs (if it is connected) or the largest component (if it does not have many separate components).* This observation verifies our conjecture regarding the property of $\lambda$ (i.e., it controls the number of effective layers

or the number of hops during the message passing aggregation schema of GNNs). Hence, the value of $\lambda$ can be searched around the diameter of the input graph.

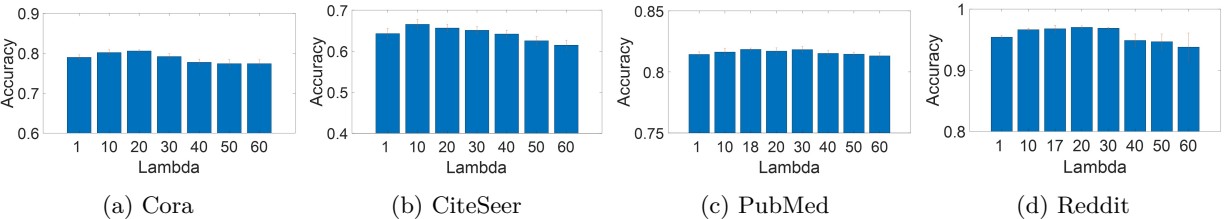

|     (a) Cora     |     (b) CiteSeer     |     (c) PubMed     |     (d) Reddit     |

Figure 6: Hyperparameter Analysis, i.e., $\lambda$ vs Node Classification Accuracy on Four Datasets.

To analyze the hyperparameter $\lambda$, we fix the feature dimension of the hidden layer to be 50, the total iteration is set to be 3000, the number of layers is set to be 60, the sampling batch size for GearGNN is 10, and GCN is chosen as the backbone model. The experiment is repeated five times for each configuration. In each sub-figure of Figure 6, the $x$-axis is the value of $\lambda$, and the $y$-axis is the accuracy of 60-layer GCN in the above setting.

First, we can observe that it's not true that GearGNN achieves the best performance with a larger $\lambda$. Specifically, we find that the optimal $\lambda = 20$ on the Cora dataset, the optimal $\lambda = 10$ on the CiteSeer dataset, the optimal $\lambda = 18$ on the PubMed dataset, and the optimal $\lambda = 20$ on the Reddit dataset. Then, natural questions to ask are *(1) what determines the optimal value of $\lambda$ in different datasets? (2) can we gather some heuristics to narrow down the hyperparameter search space to efficiently establish effective GNNs?*

Thus, we provide our discovery. In Eq. 4, we have analyzed that the decaying factor $\lambda$ controls the number of effective layers in deeper GNNs by introducing the layer-wise dependency. It means that larger $\lambda$ slows down the weight decay and gives considerably large weights to more layers such that they can be effective, and the information aggregation scope of GNN extends as more multi-hop neighbor features are collected and aggregated. In graph theory, diameter represents the scope of the graph, which is the largest value of the shortest path between any node pairs in the graph. Therefore, the optimal $\lambda$ should be restricted by the input graph, i.e., being close to the input graph diameter.

Table 5: Graph Statistics of Each Dataset.

| Metric | Cora | Citeseer | PubMed | Reddit |
|---|---|---|---|---|
| Number of Nodes | 2,708 | 3,327 | 19,717 | 4,854 |
| Connected Graph | No | No | Yes | Yes |
| Number of Components | 78 | 438 | 1 | 1 |
| Diameter of the Graph (or the Largest Component) | 19 | 28 | 18 | 17 |

Interestingly, our experiments reflect this observation. Combining the optimal $\lambda$ in Figure 6 and the diameter in Table 5, for connected graphs PubMed and Reddit, the optimal $\lambda$ is very close to the graph diameter. This also happens to Cora (even though Cora is not connected), because the number of components is not large. As for CiteSeer, the optimal $\lambda$ is less than the diameter of its largest component. A possible reason is that CiteSeer has many (i.e., 438) small components, which shrinks the information propagation scope, such that we do not need to stack many layers and we do not need to enlarge $\lambda$ to the largest diameter (i.e., 28). In general, based on the above analysis, we find the optimal value of $\lambda$ can be searched around the diameter of the input graph.

## 4 Conclusion

In this paper, we focus on building deeper graph neural networks to effectively model graph data and illustrate the oversmoothing cause from the perspective of dimensional collapse. To this end, we first provide insights regarding why ResNet is not best suited for many deeper graph neural network solutions, i.e., the *shading*

*neighbors* effect. Then, we propose a new residual architecture, Weight-decaying Graph Residual Connection (WG-ResNet), to alleviate this effect. In addition, we propose a Structure-guided Contrastive Loss (SCL) to alleviate the problem from another viewpoint, where we utilize graph topological information, pull the representations of connected node pairs closer, and push remote node pairs farther apart via contrastive learning regularization. Combining WG-ResNet with SCL, an end-to-end model GearGNN is proposed for deep graph neural networks. We provide the theoretical analysis of our proposed method and demonstrate the effectiveness of GearGNN by extensive experiment comparing with state-of-the-art methods.

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

## A    Similarity between Oversmoothing and Dimensional Collapse

In (Rusch et al., 2023), the authors describe *oversmoothing* as a phenomenon that the node representation vectors are indistinguishable from each other and thus deteriorate the performance of downstream tasks. Inspired by this, we could measure the magnitude of graph oversmoothing by the metric of covariance mean as follows:

$$covariance(h) = \frac{1}{n} \sum_i (h_i - \bar{h})(h_i - \bar{h})$$

$$\bar{h} = \frac{1}{n} \sum_i h_i$$

where $h_i$ is the node representation for node $i$ and $covariance(h) = 0$ indicates that the learned representation is indistinguishable and the deep model suffers from an oversmoothing issue. Notice that the dimensional collapse is observed when the covariance matrix of the node representations is not full-rank (i.e., the number of non-zero singular values is less than the dimension of the node representation in Figure 1). When $covariance(h) = 0$, it also indicates that $h_i = h_j = \bar{h}$ for all $i$ and $j$, and the rank of the covariance matrix of the node representation matrix is 0 (While a large value of $covariance(h)$ does not mean that the performance is ). In other words, the graph model suffers from complete collapse, where all node representations shrink to a single point. Thus, we could see that the oversmoothing issue is highly related to dimensional collapse.

## B    Visualization of the Weight of Each Layer With Different Weighting Functions

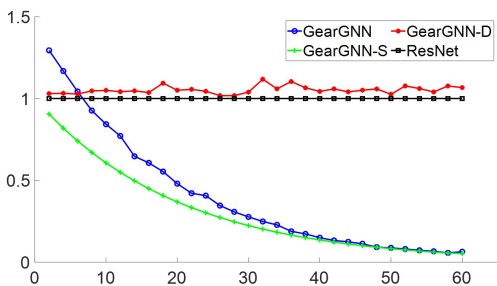

Figure 7: Weight Visualization. The $y$-axis represents the weight of each layer, and the $x$-axis represents the index of each layer, in deeper models.

Here, we visualize the weight of each layer with different weighting functions on the Cora dataset. In this experiment, we fix the feature dimension of the hidden layer to be 50; the total iteration is set to be 3000; the

number of layers is set to be 60; the sampling batch size for GearGNN is 10; GCN is chosen as the backbone model; $\lambda$ is set to be 20. In Figure 7, The $x$-axis is the index of each layer, and the $y$-axis is the weight for each layer. GearGNN-S removes the similarity measurement $e^{cos(\boldsymbol{H}^{(1)}, \tilde{\boldsymbol{H}}^{(l)})}$ in Eq. 5 and GearGNN-D removes the decaying weight factor and only keeps the exponential cosine similarity $e^{cos(\boldsymbol{H}^{(1)}, \tilde{\boldsymbol{H}}^{(l)})}$ to measure the weight for each layer. GearGNN-S achieves the simplified WG-ResNet in GearGNN, which removes the exponential cosine similarity $e^{cos(\boldsymbol{H}^{(1)}, \tilde{\boldsymbol{H}}^{(l)})}$ in GearGNN. By observation, we find that (1) ResNet sets the weight of each layer to be 1, which easily leads to *shading neighbors* effect when stacking more layers, because the faraway neighbor information becomes more dominant in the GCN information aggregation; (2) without weight decaying factor, the weight for each layer in GearGNN-D fluctuates because they are randomly independent. More specially, the weights for the last several layers (e.g., L=58 or L=60) are larger than the weights for the first several layers, which contradicts the intuition that the first several layers should be more important than the last several layers; (3) the weights for each layer in both GearGNN and GearGNN-S reduce as the number of layers increase, which suggests that both of them could address the *shading neighbors* effect to some extents; (4) combining the results from Table 4, GearGNN achieves better performance than GearGNN-S, as it imposes larger weights to the first several layers, which verifies that the learnable similarity $sim(\boldsymbol{H}^{(1)}, \tilde{\boldsymbol{H}}^{(l)})$ achieves better performance with the enlarged hypothesis space for neural networks.

## C    Proof for Proposition 2.1

**Proposition 2.1.** $\mathcal{L}_{\text{SCL}}$, based on contrastive learning, can be interpreted as the objective function of a generative adversarial network (GAN) (Goodfellow et al., 2014), which could be written as follows.

$$\min_{\theta} \mathcal{L}_{\text{SCL}} = \max_{\theta} \int_e (\tilde{P}_{pos}(e) \log(D(e)) + \tilde{P}_{neg}(e) \log(1 - D(e))) de$$

where $D(e) = f(\boldsymbol{z}_i, \boldsymbol{z}_j)$ is the discriminator of GAN with node representation $\boldsymbol{z}_i$ and $\boldsymbol{z}_j$. Also, $\tilde{P}_{pos}(e) = \sigma P_{pos}(e)$ and $\tilde{P}_{neg}(e) = \gamma P_{neg}(e)$.

*Proof.* Represent $D(e)$ by $f(\boldsymbol{z}_i, \boldsymbol{z}_j)$, then we have

$$\min_{\theta} \mathcal{L}_{\text{SCL}} = -\mathbb{E}_{v_i \in V}[\mathbb{E}_{v_j \in \mathcal{N}_i}(\sigma_{ij} \log(f(\boldsymbol{z}_i, \boldsymbol{z}_j))) + \mathbb{E}_{v_k \in \bar{\mathcal{N}}_i}(\gamma_{ik} \log(1 - f(\boldsymbol{z}_i, \boldsymbol{z}_k)))]$$

$$= -\min_{\theta} \int_e (P_{pos}(e)\sigma) \log(D(e)) + (P_{neg}(e)\gamma) \log(1 - D(e)) de$$

$$= \max_{\theta} \int_e (\tilde{P}_{pos}(e) \log(D(e)) + \tilde{P}_{neg}(e) \log(1 - D(e))) de$$

$\square$

## D    Reproducibility

All of the real-world datasets are publicly available. The experiments are performed on a Windows machine with a 16GB RTX 5000 GPU. The code of our algorithm is in an anonymous link [1]. We provide the detailed experimental setting for each experiment shown in Table 6.

Table 6: Hyperparameters for GearGNN shown in Table 2.

| Method | GearGNN |
|---|---|
| Cora | $\lambda = 20, \alpha = 0.03$ |
| CiteSeer | $\lambda = 10, \alpha = 0.02$ |
| PubMed | $\lambda = 18, \alpha = 0.1$ |
| Reddit | $\lambda = 20, \alpha = 0.02$ |

Moreover, we set the learning rate to be 0.001 and the optimizer is RMSProp, which is one variant of ADAGRAD (Duchi et al., 2011).

---

[1] https://drive.google.com/file/d/1cbNI74lhTb3LsOKhgVHT1btNz20ZLb60/view?usp=sharing

# E   Additional Effectiveness Analysis

We conduct the additional experiments by comparing our proposed method with RevGCN-Deep (Li et al., 2021) and EGNN (Zhou et al., 2021). We set the number of layers for all baseline methods to 60 for Cora, Citeseer, PubMed, and Reddit. For the OGB-arXiv dataset, we set the number of layers to 10 for all methods.

Table 7: Additional Node Classification Comparison.

| Method | Cora ($L$=60) | CiteSeer ($L$=60) | PubMed ($L$=60) | Reddit ($L$=60) | OGB-arXiv ($L$=10) |
|---|---|---|---|---|---|
| RevGCN-Deep | $0.7458 \pm 0.0084$ | $0.5137 \pm 0.0099$ | $0.8139 \pm 0.0015$ | $0.8853 \pm 0.0383$ | $0.7354 \pm 0.0009$ |
| EGNN | $0.7961 \pm 0.0036$ | $0.6566 \pm 0.0060$ | $0.8138 \pm 0.0026$ | $0.8772 \pm 0.0040$ | $0.7247 \pm 0.0015$ |
| GearGNN | $0.8059 \pm 0.0028$ | $0.6655 \pm 0.0117$ | $0.8185 \pm 0.0016$ | $0.9721 \pm 0.0011$ | $0.7401 \pm 0.0009$ |

# F   Additional Ablation Study of GearGNN

In this section, we conduct an additional ablation study to evaluate the performance of each component (i.e., the cosine similarity function in Eq. 4 and SCL formulated in Eq. 5). In Table 8, GearGNN-S denotes the variant of GearGNN after removing the cosine similarity function in Eq. 4 and SCL denotes the variant of GearGNN by removing the WG-ResNet. We have the following observations: (1). GearGNN outperforms GearGNN-S on most datasets except PubMed, which suggests that introducing the layer similarity (i.e., cosine similarity between the first layer and the $l$-th layer) can increase the performance of GearGNN(2). Compared to ContraNorm, our proposed structure-guided contrastive loss (SCL) can further boost the performance by more than 29% on average, which demonstrates the effectiveness of SCL over ContraNorm.

Table 8: Additional Ablation Study w.r.t. Node Classification Accuracy.

| Method | Cora ($L = 64$) | CiteSeer ($L = 64$) | PubMed ($L = 64$) | Reddit ($L = 64$) | OGB-arXiv ($L = 20$) |
|---|---|---|---|---|---|
| GearGNN | $\mathbf{0.8022 \pm 0.0061}$ | $\mathbf{0.6685 \pm 0.0066}$ | $0.8109 \pm 0.0033$ | $\mathbf{0.9721 \pm 0.0011}$ | $\mathbf{0.7401 \pm 0.0009}$ |
| GearGNN-S | $0.7931 \pm 0.0153$ | $0.6555 \pm 0.0085$ | $\mathbf{0.8173 \pm 0.0030}$ | $0.9621 \pm 0.0035$ | $0.7335 \pm 0.0024$ |
| ContraNorm | $0.4328 \pm 0.0320$ | $0.2128 \pm 0.0208$ | $0.3971 \pm 0.0057$ | $0.2664 \pm 0.0140$ | $0.5821 \pm 0.0324$ |
| SCL | $0.5751 \pm 0.0264$ | $0.3933 \pm 0.0076$ | $0.7601 \pm 0.0043$ | $0.9305 \pm 0.0266$ | $0.6952 \pm 0.0011$ |

# G   Experiment on the Heterophily Graph

In this section, we evaluate the performance of GearGNN on a heterophily graph (i.e., arXiv-year) from the benchmark data (Lim et al., 2021).

As shown in Table 9, our GearGNN achieves the second place performance in terms of node classification accuracy, among other state-of-the-art graph neural network methods. Interestingly, we can also observe that stacking more layers can increase the performance of GearGNN, because multi-hop neighbor information is aggregated for the message passing in heterophily graphs.

The reason why GearGNN can not achieve the best is that, although stacking layers can aggregate information from multi-hop neighbors, the loss SCL is still regulating that close neighbors should share similar representations.

Note that our design of GearGNN is not solely for heterophily graphs but for how to stack layers wisely for upgrading performance when the stacking operation is necessary and unavoidable. Hetrophily is not the only reason for stacking graph neural layers; at least, the reasoning can also originate from missing features, as shown in Section 3.2 and Table 1, where we need to stack more graph neural layers to mitigate the missing features by incorporating more neighbors.

Table 9: Node Classification on the arXiv-year dataset.

| Method | arXiv-year |
|---|---|
| GCN | $0.4602 \pm 0.0026$ |
| GAT | $0.4605 \pm 0.0051$ |
| GCNJK | $0.4628 \pm 0.0029$ |
| GATJK | $0.4580 \pm 0.0072$ |
| APPNP | $0.3815 \pm 0.0026$ |
| $H_2$GCN | $0.4909 \pm 0.0010$ |
| MixHop | $\mathbf{0.5181 \pm 0.0017}$ |
| GPR-GNN | $0.4507 \pm 0.0021$ |
| GCNII | $0.4721 \pm 0.0028$ |
| GearGNN (10 layers) | $0.4816 \pm 0.0008$ |
| GearGNN (20 layers) | $0.4835 \pm 0.0008$ |
| GearGNN (30 layers) | $0.4871 \pm 0.0007$ |
| GearGNN (50 layers) | $\underline{0.4995 \pm 0.0006}$ |

## H  Additional Hyperarameter Analysis

Here, we conduct additional hyperparameter analysis of GearGNN, i.e., $\alpha$ in the overall loss function of Eq 1.

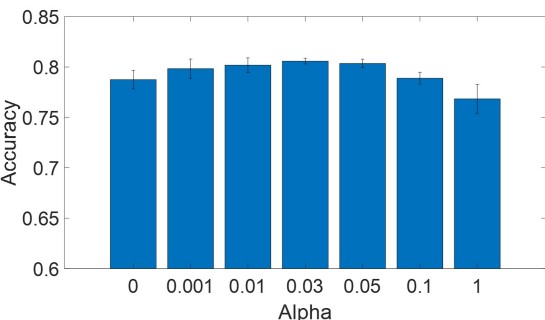

Figure 8: Hyperparameter Analysis, i.e., $\alpha$ vs Node Classification Accuracy.

To analyze the hyperparameter $\alpha$ in GearGNN, we fix the feature dimension of the hidden layer to be 50, the total iteration is set to be 3000, the number of layers is set to be 60, the sampling batch size for GearGNN is 10, GCN is chosen as the backbone model, and the dataset is Cora. We gradually increase the value of $\alpha$ and record the accuracy. The experiment is repeated five times in each setting. In Figure 8, the x-axis is $\alpha$ and the y-axis is the accuracy score. By observation, when $\alpha = 1$, the performance is worst and the performance begins to increase by decreasing the value of $\alpha$. It achieves the best accuracy when $\alpha = 0.03$. The performance starts to decrease again if we further decrease the value of $\alpha$. We conjecture that when $\alpha$ is large, it will dominate the overall objective function, thus jeopardizing the classification performance. Besides, the performance also decreases if we set the value of $\alpha$ to be a small number (i.e., $\alpha = 0.001$). In addition, comparing with the performance without using SCL regularization (i.e., $\alpha = 0$), our proposed method with $\alpha = 0.03$ can boost the performance by more than 1.8%, which demonstrates that our proposed SCL alleviates the issue of oversmoothing to some extent.

## I  Efficiency Analysis

Here, we conduct an efficiency analysis regarding our proposed method in the Cora dataset. We fix the feature dimension of the hidden layer to be 50, the total iteration is set to be 1500, the sampling batch size for GearGNN and GearGNN-S is 10, and GCN is chosen as the backbone model. We gradually increase the number of layers and record the running time.

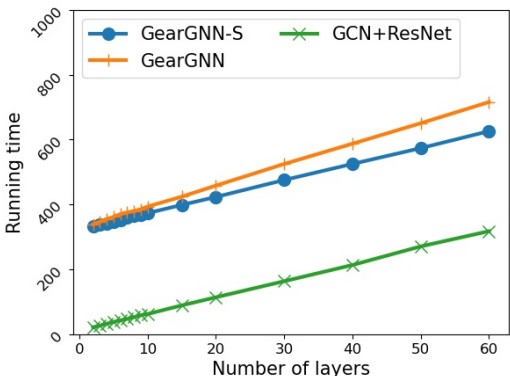

Figure 9: The Number of Layers vs Running Time (in seconds) on Cora.

In Figure 9, the $x$-axis is the number of layers and the $y$-axis is the running time in seconds. We observe that the running time of both GearGNN and GearGNN-S is linearly proportional to the number of layers. Comparing the running time of GearGNN, the running time of GearGNN-S is further reduced after the weighting function in GearGNN (e.g., $sim(\cdot)$) is replaced by a constant.

## J    Sampling Method for SCL

To realize SCL expressed in Eq. 5, we need to get the positive nodes $v_j$ and negative nodes $v_k$ towards the selected central node $v_i$. To avoid iterating over all existing nodes or randomly sampling several nodes, we propose to sample positive nodes $v_j$ and negative nodes $v_k$ from the star subgraph $S_i$ of the central node $v_i$. Moreover, to make the sampling scalable and to reduce the search space of negative nodes, we propose a batch sampling method.

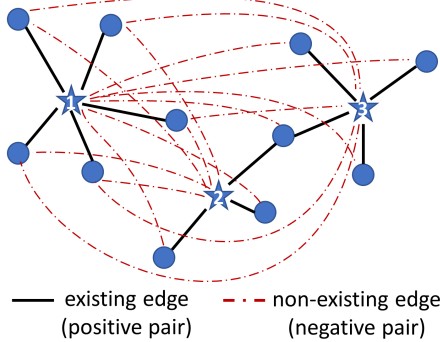

Figure 10: Batch Sampling. Each star node in the figure corresponds to node $v_i$ in Eq. 5.

As shown in Figure 10, the batch size is controlled by the number of central nodes (i.e., star nodes in the figure). For each central node, the positive nodes are those 1-hop neighbors, and the negative nodes consist of unreachable nodes. In our batch sampling, we strictly constrain that the positive nodes are only from the 1-hop neighborhood for the following three reasons: (1) they are efficient to be accessed; (2) considering all $k$-hop neighbors as positive will enlarge the scope of positive nodes and further decrease the intimacy of the directly connected nodes; (3) 1-hop positive nodes in the star subgraph can preserve enough useful information, compared to the positive nodes from the whole graph. For the third point, we prove it through the *graph influence loss* (Huang & Zitnik, 2020) in Proposition J.1, and the formal definition of *graph influence loss* is given in the following paragraph after Proposition J.1.

**Proposition J.1** (Bounded Graph Influence Loss for Sampling Positive Pairs Locally)**.** *Taking GCN as an example of GNN, the graph influence loss $R(v_c)$ on node $v_c$ w.r.t* **positive nodes from the whole graph** *against* **positive nodes from the 1-hop neighborhood star subgraph** *is bounded by $R(v_c) \leq (n - d_c)\frac{\mu}{(D_{GM}^{\bar{\mathcal{P}}_*})^{|\bar{\mathcal{P}}_*|}}$ , where $n$ is the number of nodes, $d_c$ is the degree of node $v_c$ including the self-loop, $\mu$ is a constant, $\bar{\mathcal{P}}_*$ is the path from center node $v_c$ to a 1-hop outside node $v_s$ which has the maximal node influence $I_{v_c,v_s}$, and $|\bar{\mathcal{P}}_*|$ denotes the number of nodes in path $\bar{\mathcal{P}}_*$.*

*Proof.* According to the assumption of (Wang & Leskovec, 2020), $\sigma(\cdot)$ can be identity function and $\boldsymbol{W}^{(\cdot)}$ can be identity matrix. Then, the hidden node representation (of node $v_c$) in the last layer of GCN can be written as follows.

$$\boldsymbol{h}_c^{(\infty)} = \frac{1}{D_{c,c}} \sum_{v_i \in \mathcal{N}_c} A_{c,i} \boldsymbol{h}_i^{(\infty)}$$

Then, based on the above equation, we can iteratively replace $\boldsymbol{h}_i^{(\infty)}$ with its neighbors until the representation $\boldsymbol{h}_s^{(\infty)}$ of node $v_s$ is included. The extension procedure is written as follows.

$$\boldsymbol{h}_c^{(\infty)} = \frac{1}{D_{c,c}} \sum_{v_i \in \mathcal{N}_c} A_{c,i} \frac{1}{D_{i,i}} \sum_{v_j \in \mathcal{N}_i} A_{i,j} \cdots$$
$$\frac{1}{D_{k,k}} \sum_{v_s \in \mathcal{N}_k} A_{k,s} \boldsymbol{h}_s^{(\infty)}$$

The above equation suggests that the influence from the positive node $v_s$ to the center node $v_c$ is through the path $\mathcal{P} = (v_c, v_i, v_j, \ldots, v_k, v_s)$.

Following the above path formation and assuming the edge weight $A(i,j)$ as the positive constant, according to (Huang & Zitnik, 2020), we can obtain the node influence $I_{v_c,v_s}$ of $v_s$ on $v_c$ as follows.

$$I_{v_c,v_s} = \|\partial \boldsymbol{h}_c^{(\infty)} / \partial \boldsymbol{h}_s^{(\infty)}\| \leq \frac{\mu}{(D_{GM}^{\bar{\mathcal{P}}})^{|\bar{\mathcal{P}}|}}$$

where $\mu$ is a constant, $D_{GM}^{\bar{\mathcal{P}}}$ is the geometric mean of the degree of nodes sitting in path $\bar{\mathcal{P}}$, and $\bar{\mathcal{P}}$ is the path from the positive node $v_s$ to the center node $v_c$ that could generate the maximal multiplication of normalized edge weight, $|\bar{\mathcal{P}}|$ denotes the number of nodes in path $\bar{\mathcal{P}}$.

The above analysis suggests that the node influence of positive long-distance nodes is decaying.

Hence, the graph influence loss about learning node $v_c$ from **the whole graph positive nodes** versus **from the 1-hop localized positive nodes** can be expressed as follows.

$$I_G(v_c) - I_L(v_c) = I_{v_c,v_1} + I_{v_c,v_2} + \ldots + I_{v_c,v_{n-d_c}}$$
$$\leq \sum_{i=1}^{n-d_c} \frac{\mu_i}{(D_{GM}^{\bar{\mathcal{P}}_i})^{|\bar{\mathcal{P}}_i|}}$$
$$\leq (n - d_c)\frac{\mu_*}{(D_{GM}^{\bar{\mathcal{P}}_*})^{|\bar{\mathcal{P}}_*|}}$$

where $I_G(v_c)$ denotes global influence, $I_L(v_c)$ is the influence for star subgraph, $d_c$ is the degree of node $v_c$ (including self-loop), and $\frac{\mu_*}{(D_{GM}^{\bar{\mathcal{P}}_*})^{|\bar{\mathcal{P}}_*|}}$ is the maximal among all $\frac{\mu_i}{(D_{GM}^{\bar{\mathcal{P}}_i})^{|\bar{\mathcal{P}}_i|}}$. $\qquad\square$

Specifically, the graph influence loss (Huang & Zitnik, 2020) $R(v_c)$ can be expressed as $R(v_c) = I_G(v_c) - I_L(v_c)$, which is determined by the global graph influence on $v_c$ (i.e., $I_G(v_c)$) and the star subgraph influence on $v_c$ (i.e., $I_L(v_c)$). Then, to compute the graph influence $I_G(v_c)$, we need to compute the node influence of each node $v_j$ to node $v_c$, where node $v_j$ is reachable from node $v_c$. Based on the final output node representation vectors, the node influence is expressed as $I_{v_c,v_j} = \|\partial \boldsymbol{h}_c^{(\infty)} / \partial \boldsymbol{h}_j^{(\infty)}\|$, and the norm can be any subordinate norm (Wang & Leskovec, 2020). Then, $I_G(v_c)$ is computed by the $L1$-norm of the following vector, i.e.,

$I_G(v_c) = \|[I_{v_c,v_1}, I_{v_c,v_2}, \ldots, I_{v_c,v_n}]\|_1$. Similarly, we can compute the star subgraph influence $I_L(v_c)$ on node $v_c$. The only difference is that we collect each reachable node $v_j$ in the star subgraph $L$ (i.e., 1-hop neighbors of $v_c$). Overall, in Proposition J.1, we show why positive pairs can be locally sampled with the support from graph influence loss of a node representation vector output by the GCN final layer.

