# OpenReview forum: "Revisiting Residual Connections for Neural Structure Learning"
_TMLR — Rejected by TMLR_

### Review · Reviewer_TLCw · 2024-02-06

**Summary Of Contributions:**

This paper first verifies that normal GCNs with residual connections have a rank deficiency problem, that is, the fast decay of singular values of activations. Also, this paper claims that the information collected from faraway neighbors becomes dominant by introducing the usual residual connections. In order to address these issues, the paper proposes GearGNN, which combines Weighted-Decaying Graph Residual Connection (WG-ResNet) and Structured-Guided Contrastive Loss (SCL). Theoretical analyses show that GearGNN's loss function is equivalent to adversarial training for distinguishing positive and negative edges. GearGNN is applied to node classification tasks on real datasets and verified to solve problems of dimensional collapse.

**Audience:**

Yes

**Broader Impact Concerns:**

N.A.

**Claims And Evidence:**

No

**Requested Changes:**

1. I have a question about the description of residual connections in Eq. (2). It should be $\boldsymbol{H}^{(l)} = \sigma(\hat{A}\boldsymbol{H}^{(l-1)}\boldsymbol{W}^{(l-1)}) + \boldsymbol{H}^{(l-1)}$, not $+ \boldsymbol{H}^{(l-2)}$. Since the proposed method (specifically, Eq. (4) of WG-ResNet) is based on this formulation, it needs to be modified.

2. More detailed explanations of Eq. (5) are needed, specifically, the design of $\sigma_{ij}$'s and $\gamma_{ik}$'s. Looking at the proof of Theorem 2.2, there is no restriction on $\sigma_{ij}$ or $\gamma_{ik}$ as long as it represents probability distributions. Therefore, we cannot find any necessity to set them as in Eq. (5) from the theoretical analysis. I want to clarify the rationale of their design.

3. There is room for improvement in the use of the formulas, making it difficult to follow the equations to see if the proofs are mathematically rigorous and undermine the paper's clarity. For example, $P_{pos}$, $P_{neg}$, and $\sigma$ are undefined, and the relationship between $P$ and $P_{pos}$ and $P_{neg}$ and between $\sigma$ and $\sigma_{ij}$ are not explicitly shown. My understanding is that $\sigma$ is a value that depends on $e$, so if $e=(i, j)$, then $\sigma$ should be written as $\sigma_{ij}$. Regarding, $P$, looking at the proof in Theorem 2.2, $P_{pos}(e)$ equals to $P(y=1)$. However, I have two comments about it. First, $y$ is not defined explicitly. Second, since $P_{pos}(e)$ is the probability that a particular positive edge $e$ is chosen, while $P(y=1)$ is the probability that the chosen edge is in the positive edge set. So, they should represent different concepts.

4. Theorem 2.2 claims that if the model is an optimal discriminator, then this model can address the over-smoothing. However, I have two questions about this. First, since this paper does not mathematically define what "address the over-smoothing" means, it is unclear what Theorem 2.2 claims. Second, over-smoothing is an issue of model representativeness, not optimization. Over-smoothing is not caused by the inappropriate choice of the loss function but by the inadequate expressive power of the model to achieve the optimum. Actually, since ordinal GCNs also use the loss functions designed to discriminate nodes (typically, the cross entropy loss), if the optimal is achieved, over-smoothing can be avoided, similarly to the proposed method. Therefore, Theorem 2.2 is too strong an assumption to claim that the proposed model avoids over-smoothing.

5. The dataset used in the experiments is a graph with high homophily, and its applicability to the task of node classification on a graph with high heterophily has not been tested.

6. In the text (e.g., the upper part of P4), $\hat{\boldsymbol{A}}$ is not italicized, while in the formulas (e.g., Eq. (3)), it is. The fonts should be unified.

**Strengths And Weaknesses:**

Strengths
1. In numerical experiments, increasing the number of layers (up to 64) on various datasets consistently improves accuracy, alleviating the rank deficiency problem.
2. The proposed method is model-agnostic and is numerically evaluated using at least three types of GNNs (GCN, GAT, and GraphSAGE).
3. The ablation study shows that each part of the proposed method contributes to performance improvement.

Weaknesses
1. There is room for improvement in mathematical descriptions (e.g., proof of Theorem 2.2 and the description of residual connections.)
2. The theoretical necessity of the proposed method (especially the design of $\sigma_{ij}$'s and $\gamma_{ik}$'s) is questionable.
3. The assumptions of the theoretical analysis (Theorem 2.2) are too strong to claim that the proposed method avoids over-smoothing.
4. Numerical experiments have only been performed on homophily datasets. The prediction performance on heterophilic graph datasets is not tested.

---

### Review · Reviewer_CCcK · 2024-02-06

**Summary Of Contributions:**

The paper suggests that graph neural networks regularly suffer from dimensional collapse in their learned representations and that this has a negative impact on performance.  To address this, the authors suggest GearGNN, which adds a regularization loss called SCL, an additional contrastive loss term, and propose a modified residual connection they call WG-ResNet that weights the non-residual path by the similarity between the new features and the original input features at a node.

They claim these improvements improve performance while avoiding dimensional collapse in the learned representations.  In terms of empirical results they demonstrate performance on some baselines (Cora, CiteSeer, PubMed, Reddit and OGB-arXiv).

**Audience:**

Yes

**Claims And Evidence:**

No

**Requested Changes:**

Please clarify Sections 2.3 and 2.2.  As written I don't think the method is reproducible.  The equations are two ambiguous.  Consider adding some python code or something to disambiguate what is actually done.

Please clarify whether the results in Section 3 were generated by the authors or from previously reported results.

Please look at the effect the ablations have on the singular value spectrum, without this I feel like the central story of the paper is missing its middle.

Please do grammar pass on the draft, there are number of grammatical errors in the text.

**Strengths And Weaknesses:**

At the start, I take some issues with the premise of the paper.  Why do we care if the learned representations are lower dimensional than we maybe intended?  Even low dimensional representations can be very informative, geometry isn't the same as information.  Yet, the paper takes as an unstated axiom that having flatter singular value spectra is better. If this were the case, if we simply whitened the learned representations, wouldn't they improve? This is not my experience.

In equation (2) why is the residual connection written as bringing forward the representation from *two* layers back? This notation continues on the next page.

I disagree with the premise of Section 2.2.  Maybe I misunderstood the argument, but the text at the start of the section seems to be trying to argue that as we go deeper in a graph neural network, because that is akin to increasing the receptive field for each node, that this means that a graph neural network upweights information from nodes farther away.  I don't think this is correct.

Consider a simple example of a square lattice where in each layer each node just adds its own value (a residual connection) that that of its four neighbors north south east and west.  If we do that for three layers, the weight a node would have for those below it take this pattern:

```
 0   0   0   1   0   0   0
 0   0   3   3   3   0   0
 0   3   6   12  6   3   0
 1   3   12  13  12  3   1
 0   3   6   12  6   3   0
 0   0   3   3   3   0   0
 0   0   0   1   0   0   0
```

While the receptive field has grown, we continue to accumulate more weight from the central node as time goes on.  So, if the claim is that a standard graph neural network, as we increase in depth places larger weight on nodes farther from the current node, I disagree.  I don't understand why the first two terms are labeled as being "information aggregated from faraway neighbors" in equation 3.  What are labeled are simply the features of the previous two layers, but local to the node in question.

The notation of equation (4) is hard to parse, please consider either a less ambiguous notation or including some pseudocode that disambiguates the intention.  For instance, its not necessarily clear whether the intention in equation 4 is to compute the similarity per dimension or not.

Am I to take equation (4) at its word, is the similarity always computed with respect to the input features $H^(1)$?  Wouldn't the effect of this term just be to keep the representations close to their input representations?  This will surely flatten the singular value spectrum (assuming the input has a relatively flat spectrum), but it also very clearly keeps the network from doing much of anything.  Also, its not clear what would happen if the network had different widths at different depths (or even a different width than the input dimensionality).

Did the authors implement all of the baseline results reported in section 3? It appears as though these are just copied from previous work.  At which point I don't really think we can compare these honestly.  Perhaps the improvements shown are simply due to longer training, a better optimizer, better hyparameter settings or other issues of "graduate student descent".

In that regard, I find the ablation results in Section 3.4 as better evidence. Here at least I'm reasonably sure these are fair comparisons to make.  It does appear to support the idea that the proposals made in the paper improve performance, everything else held equal.  However, I feel here the paper falls short of connecting to its earlier story.  Is the increase in performance because these two additions actually flatten the eigenspetrum.  Granted, the paper does show in Figure 4 that there are few zero singular values in their learned representations, but these are not ablative comparisons, these are comparisons to prior work if I understand correctly.  At which point, once again I'm less sure the improvements are for the *stated* reasons.  I'll also point out that the dominance of the GearGNN in Figure 4 isn't so commanding.  It seems like GCNII and ContraNorm show similar number of non-zero singular values.

Section 3.5 attempts to show that the GearGNN improvements can improve different baselines.  This is also a better form of evidence in my view than the results presented in Section 3.3, but I'd like to ask directly if the GAT and GraphSage numbers are the authors own implementation or rather the previously reported results.  I'll also point out that directly below Figure 5, the paper says "By observation, we find that both GAT and GraphSage suffere from oversmoothing", but I'd like to point out that Figure 5 doesn't actually show that. Figure 5 shows that the performance is worse, and there are some missing steps to go from that to an argument that the reason it has worse performance is oversmoothing.  In general, its this kind of implied logical leaps that I feel like the paper is guilty of more broadly.


I don't really understand what the point of Section 3.2 or 3.6 are? What role do they play in the central story of the paper?  The best I can come up with is that its an attempt to try to explain which datasets require fundamentally deep GCNs to solve, as the authors believe the problem of dimensional collapse only gets worse with depth.  I'll point out though that these are all undemonstrated assertions.

---

### Review · Reviewer_aoSm · 2024-02-12

**Summary Of Contributions:**

The paper introduces a Weight-Decaying Graph Residual Connection (WG-ResNet), and a Structure-Guided Contrastive Loss (SCL) to tackle the over-smoothing problem in deep Graph Neural Network (GNN) models. Through extensive experiments, the combined use of WG-ResNet and SCL is demonstrated to outperform some baseline methods.

**Audience:**

Yes

**Broader Impact Concerns:**

No concerns

**Claims And Evidence:**

No

**Requested Changes:**

Please fix issues in your lemma and theorem. If there is not a clear statement, a discussion is a better way to state the results.

Increase the clarity of the writing.

**Strengths And Weaknesses:**

**Strengths**:

* Structure-guided contrastive learning incorporates topology information into contrastive loss, promoting the similarity of nodes with similar topological structures.
* Experimental results show better performance of the proposed method, particularly with deeper GNNs. The ablation study confirms the effectiveness of each component.

**Weaknesses**:
* Ambiguous statements in theorems. *Every statement* in the theorem must be rigorously defined. In Lemma 2.1, what is "a generative adversarial network"? What is the definition of GAN-based constrastive learning loss?  In theorem 2.2, what is the definition of "the oversmoothing problem"? What does "address" mean?

* Ambiguous motivation for similarity measure in weight decaying. The choice of using the cosine similarity between each layer and the first layer feature for weight decaying is not well-explained. This may lead to learned features closely resembling initial features, hindering the learning of new features.

* Lack of clarity in the motivation for weight decaying. The exponential decay function's motivation is unclear, making deeper features less important while preserving shallower features in residual connections. How about adaptive/learnable weight decay function?

* Insufficient experimental results. The paper proposes two approaches to address over-smoothing but does not clearly demonstrate the individual effectiveness of each component. Comparative experiments, such as WG-ResNet vs. ResNet GNN and SCL vs. vanilla contrastive GNN (ContraNorm), are needed to demonstrate the effectiveness of each component.

* Typos in the paper, such as "excursively" on page 2.

---

### Decision · Action_Editor_qRAC · 2024-03-23

**Recommendation:** Reject

**Comment:**

I summarize below the recommendations emerged during review. They mostly have to do with needing more precision.

 - The main invoked motivation is alleviating over-smoothing, but the applicability of the theoretical analysis is something reviewers and authors did not reach consensus on in the discussion period: if over-smoothing is related to insufficient expressive power but the theoretical arguments assume sufficient power or convergence to the optimum, more substantial discussion of this gap is needed.
 - The premise that lower dimensional representations mean less information needs to be questioned and discussed more seriously. The response cites some past work on dimensional collapse, but this should be related to the premise of this paper and perhaps contextualized.
 - The argument in section 2.2 that "information collected from faraway neighbors becomes dominant" is not sufficiently convincing: a sum of more terms isn't necessarily larger than a sum of (different) fewer terms, without further structure.
 - Proposition 2.2 lacks sufficiently rigorous wording, making it hard to discuss whether the proof proves it.
 - There is a gap between proposition 2.2 and the form of $\sigma_{ij}$ and $\gamma_{ij}$ given in Eq. (5) and it not clear where the form given comes from or whether it is theoretically justified/necessary.
 - Appendix F makes too strong claims ("indispensable") for relatively small differences.

**Audience:**

Graph representation learning is a topic of interest for the TMLR audience, and reviewers agree that a better supported version of this work would be relevant.

**Claims And Evidence:**

During the review and discussion period, the reviewers have raised several concerns with the rigorousness and clarity of some of the core motivating claims in this paper: the relationship to over-smoothing, the motivation for decay as rearrangement of terms in a sum (2.2), and the assumption that low-dimensional representations are insufficient.

In addition, the reviewers have pointed out some overall issues that overall decrease faith in the claims: reviewers are generally unsatisfied with the precision of the more mathematical parts.

Given the TMLR criteria, these fundamental issues are important enough to recommend a resubmission after a more substantial revision. We appreciate the revisions, but to reach a satisfactory threshold of accuracy, more than just wording changes are necessary. We encourage the authors to focus on the pain points identified by reviewers, and either strengthen the evidence, or amend the claims if necessary. I add that theorizing is not the only way to support claims, but if you choose to support claims in a theoretical fashion, then more precision is required.

**Resubmission Of Major Revision:**

The authors may consider submitting a major revision at a later time.